# Prevalence of undiagnosed diabetes mellitus and associated factors among adult residents of Mizan Aman town, Southwest Ethiopia: Community-based cross-sectional study

Tsegaye Atrese[1], Lata Fekadu[2], Guta Kune[2]*, Abel Shita[1], Kifle Woldemikael[2]

1 Department of Public Health, Mizan Aman Health Science College, Mizan Aman, South West Ethiopia,
2 Department of Epidemiology, College of Public Health, Jimma University, Jimma, South West Ethiopia

* gutakune12@gmail.com

## Abstract

### Background

Diabetes mellitus continues to be a significant global public health concern, and it is currently a public health issue in developing nations. In Ethiopia, about three fourth of adult population with diabetes are unaware of their diabetic condition. However, there is a limited research on this specific topic particularly in the study area.

### Objective

To assess prevalence of undiagnosed diabetes mellitus and its associated factor among adult residents of Mizan Aman town, south West Ethiopia.

### Methods and material

A community-based cross-sectional study was conducted from May 23 to July 7, 2022, on 627 adult residents of Mizan Aman town. A multi stage sampling technique was used to obtain 646 study units. Interviewer-administered structured questionnaires were employed to gather socio-demographic and behavioral data. Anthropometric measurements were obtained and blood samples were taken from each participants. The fasting blood glucose level was measured after an 8-hour gap following a meal, using a digital glucometer to analyze a blood sample. Data were cleaned and entered into Epi-data v 3.1 and exported to SPSS v. 26 for analysis. Bi-variable analysis was done to select candidate variables and multivariable logistic regression model was fitted to identify independent predictors of undiagnosed diabetes mellitus. Adjusted odds ratio (AOR) with 95% CI was computed and variables with p-value < 0.05 were declared to be predictors of undiagnosed diabetes mellitus.

### Results

The study revealed that, the overall magnitude of undiagnosed diabetes mellitus was 8.13% (95% CI: 6.1, 10.6). Predictors of undiagnosed diabetes mellitus were; physical activity level less than 600 Metabolic equivalent/min per week (AOR = 3.39, 95%CI 1.08 to 10.66), family

**Data Availability Statement:** All relevant data are within the paper and its Supporting information files.

**Funding:** The author(s) received no specific funding for this work.

**Competing interests:** The authors have declared that no competing interests exist.

**Abbreviations:** AOR, Adjusted Odds Ratio; BMI, Body Mass Index; COR, Crude Odds Ratio; CSA, Central Statistical Agency; DM, Diabetes Mellitus; IDF, International Diabetic Federation; SPSS, Statistical Package for Social Science; SSA, Sub-Saharan Africa; UDM, Undiagnosed Diabetes Mellitus; WC, Waist Circumference; WHO, World Health Organization.

history of diabetes mellitus (AOR = 2.87, 95% CI 1.41, 5.85), current hypertension(AOR = 2.9, 95% CI 1.26, 6.69), fruit consumption of fewer than three servings per week(AOR = 2.64, 95% CI 1.18 to 5.92), and sedentary life(AOR = 3.33, 95% CI 1.63 to 6.79).

## Conclusion

The prevalence of undiagnosed diabetes mellitus was 8.13%. Physical inactivity, family history of diabetes mellitus, current hypertension, sedentary life, and fruit servings fewer than three per week were independent predictors of undiagnosed diabetes mellitus.

## Introduction

Diabetes mellitus is a group of metabolic diseases characterized by hyperglycemia resulting from defects in insulin secretion, insulin action, or both [1]. Undiagnosed diabetes mellitus (UDM) is a condition in which a person has not been diagnosed with diabetes mellitus and whose blood glucose level currently meets criteria established for the diagnosis of diabetes mellitus [2]. Undiagnosed diabetes mellitus can only be identified through a health survey, which tests a person's blood sugar levels and asks about a history of being diagnosed with diabetes mellitus [3].

Diabetes mellitus remains a major public health problem worldwide, regardless of a country's level of development and income. Undiagnosed DM is a challenge even in developed countries, even with universal access to healthcare and advanced healthcare technologies [4]. Globally, approximately 537 million adults between the ages of 20 and 79 will be living with DM in 2021. However, approximately half (44.7%) of them were unaware of their status [5]. Interestingly, 75% of adults who were unaware of their diabetes status are from low- and middle-income countries (LMIC) [4, 6]. In recent decades, the incidence of diabetes mellitus in Sub-Saharan Africa (SSA) has increased rapidly as a result of globalization, rapid urbanization and lifestyle changes; increased consumption of unhealthy diets and sedentary lifestyles [7].

According to a report published by the International Diabetic Federation in 2021, the proportion of undiagnosed diabetes in SSA was 54%, a 4% increase from 2019, with the African region having the highest prevalence of when compared other regions [5]. In the same year, the estimated mortality associated with DM in the African region was approximately 416,000, due to late diagnosis and failure to seek treatment [6, 8]. On the other hand, North America and the Caribbean region has the lowest proportion (37.8%) globally [8]. In Ethiopia, among populations living with diabetes, according to IDF 2017 estimate; approximately 1.96 million (76%) of them are undiagnosed and unaware of their diabetes status [9].

Besides, DM has been the leading cause of morbidity and mortality in association with numerous complications, like blindness, kidney failure, heart attacks, stroke and lower limb amputation [10, 11].

In addition to the health burden, diabetes also impacts the individual's economy, healthcare system and government spending by incurring high medical costs. Estimated global healthcare spending related to diabetes was approximately $376 billion in 2010, accounting for approximately 12% of total healthcare spending. For these UDM reasons, an additional cost of $2864 per year per person was associated with complications related to late diagnosis and treatment [12–14]. In Ethiopia, the estimated cost of NCDs is at least 31.3 billion Ethiopian birr (US$1.1 billion) per year, equivalent to about 1.8% of gross domestic product (GDP) [2]. To reduce the health and economic burden of non-communicable diseases, including diabetes mellitus, the

United Nations set global goal for the reduction and prevention of non-communicable diseases, which is included in the Sustainable Development Goal(SDG) under Target 3.4, which calls for 30% reduction in premature deaths from non-communicable diseases, including diabetes mellitus, by the end of 203 [15]. To achieve this plan, global country-specific strategies have been adopted by WHO, such as improving screening, early detection and treatment for non-communicable diseases and integrating service delivery to existing health system [15–19].

The Ethiopian Federal Minister of Health (FMOH) established various strategies such as: reducing key modifiable risk factors, developing and implementing legal framework, aligning the healthcare system towards person-centered care, and promoting quality research to avert the burden of NCDs in Ethiopia [20]. Despite these strategies and the availability of diagnostic screening tools, the majority of individuals come to healthcare after the disease has been progressed and multiple organ damage has occurred [21].

Factors such as educational level, body mass index, family history and the presence of other chronic illnesses have been associated with undiagnosed DM [22]. However, studies conducted in some areas of Ethiopia show varying evidence of the extent and associated factors of undiagnosed diabetes, with the extent of UDM varying between 11.5% and 2.3% [23, 24]. Apart from that, some of the studies lack information on some variables like physical activity and it is recommended to do more studies on such variables. Similarly, some of the studies also used small sample size, compromising the generalizability of the study results [25]. In addition, the evidence on undiagnosed diabetes mellitus and associated factors was limited in the study area, and some of the studies conducted on diabetes mellitus were also outdated. The results of this study will inform policymakers in developing strategies and prioritizing resource allocation, and improve community-based screening programs for early detection and improve interventions for the prevention and control of undiagnosed diabetes mellitus.

## Methods and material

### Study design, study area and period

A community-based cross-sectional study was conducted in Mizan-Aman town from May 23 to July 7, 2022. Mizan Aman town is the administrative center of Bench-Sheko zone, Southwest Ethiopia. The town is located 564 km from Addis Ababa, the capital city of Ethiopia. The town has an estimated population of 84,827 based on the 2007 Ethiopian Central statistical Agency projection. It has five kebeles (the least administrative unit). Regarding health facilities; currently, there are 72 governmental and private health facilities which are serving the community [26].

### Study participants

All adult residents of the study area age 18 years and above were eligible to participate in the study. All adults of the selected households who fulfilled the inclusion criteria were the study populations. A randomly selected household of the selected kebele was the sampling unit and an adult age 18 years or above from a randomly selected household who actually participated in the study was the study unit. Age over 18 years old, who have lived in the study area for 6 months or more and have not been previously diagnosed with diabetes mellitus were included in the study. Critically ill individuals who were unable to communicate and respond to the questions, pregnant women and adults who were taking beta blockers within two weeks of data collection were excluded from the study.

## Sample size determination and sampling technique

The sample size was determined by using single population proportion formula by using Epi info, considering the following parameter; prevalence of undiagnosed diabetes mellitus 10.2% from previous study [27], 95% confidence level, 3% margin of error, 1.5 design effect, and by adding 10% non-response rate. Then, the final sample size was 646. A multistage sampling technique was used and two kebeles (Hibret and Addis-Ketema) were randomly selected from the five kebeles in Mizan-Aman town. The sample size for each selected kebele was proportionally allocated based on the size of households in each kebele. House-holds were then identified by using a systematic random sampling technique after the sampling interval. Finally, for households with more than one eligible participant, the lottery method was used to select one study participant among those eligible household members. If a household doesn't have any eligible residents during the initial visit, three revisits were conducted. If they were still unavailable during the three revisits, the next nearby household was chosen as a substitute. After collecting data from the substituted household, the original interval was maintained to select the next household (Fig 1).

## Data collection tools and procedure

Data was collected by four diploma nurses who were experienced in data collection procedures using a pre-tested structured interviewer-administered questionnaire. The tool is adapted from Ethiopia's steps report on risk factors for non-communicable diseases and the prevalence

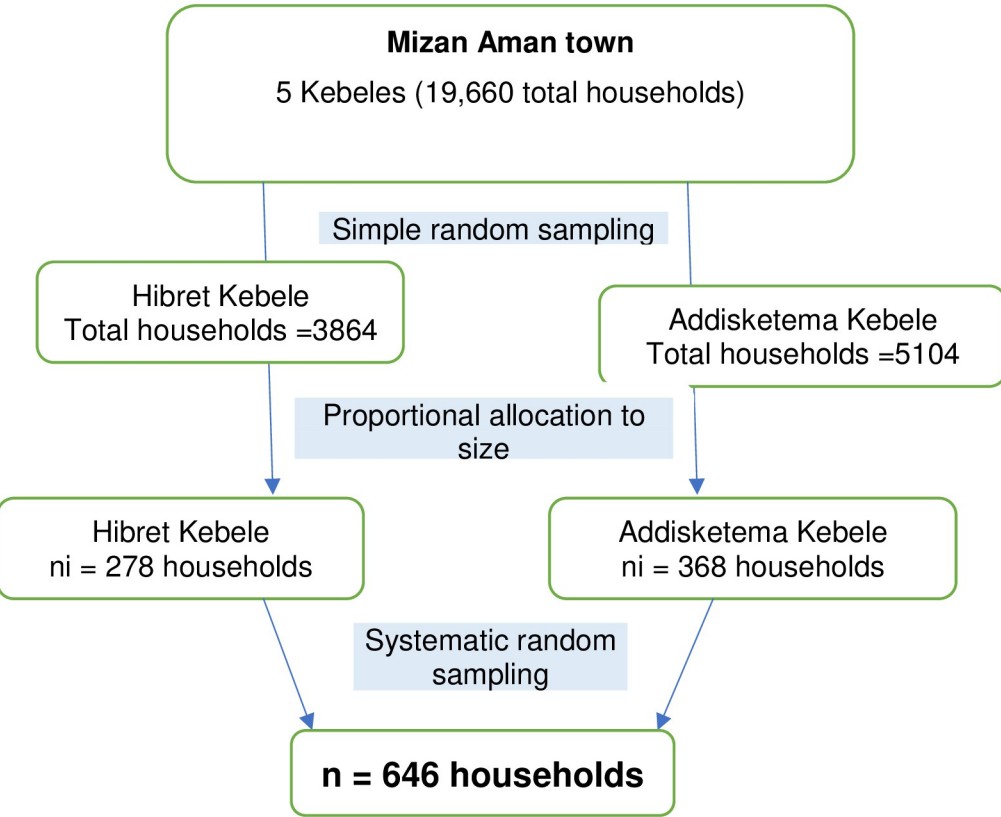

**Fig 1. Sampling procedure to study prevalence of undiagnosed diabetes mellitus and associated factors among adult residents of Mizan Aman town, 2022.**

of selected NCDs [28]. The questionnaire contains socio-demographic, behavioral, and biological factors. The data collection tool was initially prepared in English and translated to Amharic and then translated back to English for its consistency.

Socio-demographic and behavioral related data were collected through an interviewer administered structured questionnaire while biological related questions were collected by using pre-tested instruments. Seca mechanical weight scale was used to measure the weight, tap meter was used to measure height and waist circumferences, and CareSens™ N Eco digital glucometer was used to measure fasting blood glucose level of the participants. The activities were supervised by two health officers and the principal investigator throughout the data collection process.

Anthropometric measurements like weight is measured by using Seca-mechanical weighting scale to the nearest 0.1 kg after calibration, height measured in a standing position in centimeters to the nearest 1 centimeter (cm). Then body mass index (BMI) was calculated by dividing the weight by height in meter squares ($h^2$). Then BMI $\geq$25 kg/m$^2$ was classified as high and <25 kg/m$^2$ were classified as normal [29].

Meanwhile, waist circumference was measured at a level midway between the lower rib margin and the iliac crest on the horizontal plane around the body. Then WC $\geq$94 cm for males and WC $\geq$80 cm for females was classified as high and <94 for males and <80 for females was classified as normal [30].

Regarding questionnaires related to dietary assessment, we prepared nutrition card that show of the examples of local fruits and vegetables. Each picture represents the size of servings and we asked them to answer how many days they eat fruit or vegetables in particular week, how many servings of fruit or vegetables they consume. And also we asked type of oil/fat mostly used for meal preparation.

To measure blood pressure participants were asked to sit and relax for 30 minutes before the measurement, avoid crossing their legs, keep their feet on floor, support their back, ensure an empty bladder, refrain from smoking, and abstain from consuming coffee or tea. Then the measurement was done while the participant was seated, using a digital blood pressure device on the non-dominant arm. Three readings were taken with five minutes interval between each reading, and the average of the second and the third reading was used to determine the participants' blood pressure level.

In order to measure glucose level, participants were instructed to be on overnight fasting for at least 8 hours. Then, in the early morning of the next day; a blood sample was collected from each participant from their middle or ring finger, using CareSens™ N Eco digital glucometer to measure blood glucose level. Accordingly, participants who had fasting blood glucose levels of 126mg/dl or above were classified as having undiagnosed DM and those who had blood glucose measurements of 100 -125mg/dl were classified as impaired fasting glucose and those with FBS levels less than 100 mg/dl were classified as normal for blood sugar level [31]. Furthermore, participants who were diagnosed to have diabetes were informed about their disease status and were advised to visit nearby health facility for further treatment.

On the other hand, participants who were found to have impaired fasting glucose (IFG) levels and normal blood glucose levels were also counseled on risk factors of NCDs like physical inactivity, smoking, and alcohol drinking and advised to visit health care facilities for further checkup. Similarly, participants with blood glucose measurement levels <70 mg/dl were advised to have meals containing carbohydrates, fruit, and juice.

One day training was given for data collectors and supervisors on study objectives, data confidentiality, how to take blood sample and test, how to use the data collection instruments, the contents of the questionnaire in detail, and the data collection method by the principal investigator. Demonstration on measurements was conducted during training session and the

trainees re-demonstrated the procedure of sample collection and physical measurements. Then, pretest was conducted on 32 (5%) of the sample in Kometa kebele. After the pretest had been conducted the results were reviewed by the principal investigator together with the data collectors and supervisors. Subsequently, some questions which were difficult or not easily understandable during the pretest were re-written in a way that the participants can easily understand and the flow of the questions was also revised and modification to the skip rule was made. The completed questionnaire was checked daily by the principal investigator for possible errors and some incomplete data were refilled by the data collectors during the next day visit for blood specimen.

## Operational definition

**Undiagnosed diabetes mellitus**- is a condition in which participants fasting blood glucose level was 126mg/dl and above for the first time.

**Physical activity**- is any bodily movement produced by skeletal muscle that results in energy expenditure.

**High-level physical activity**: is when the calculated MET-min/week for all combinations of activities is greater or equal to 3000 MET-min per week.

**Moderate level physical activity**: is when the calculated MET-min/week for all combinations of activities is between 600 to 2999 MET-min per week.

**Low-level physical activity**: is when the calculated MET-min per week is less than 600.

**Sedentary behavior**: the time spent sitting at home traveling in a car, watching television, using social media, and sitting at office work.

**Hypertension**: an individual's blood pressure measurement $> 140/90$ mmHg.

**The metabolic equivalent of task-minute per week**- the number of days an individual performs an activity times the time expended multiplied by the constant value for the intensity level of activity.

**Fasting blood glucose level (FBS)**- is the level of blood glucose measured after overnight fasting.

**Current hypertension** is an individual's blood pressure measurement $\geq 140/90$ mmHg in which was not previously diagnosed as hypertension.

## Data analysis

The data was reviewed, coded and entered into Epi-Data v.3.1and analyzed by SPSS v.26. Data were described in terms of frequency, mean and percentage, and then presented in table and graph form. Multi-collinearity between independent variables was checked using variance inflation factor. However, no significant multi-collinearity was detected. The Hosmer-Lemeshow test of goodness fit was done to check the suitability of the model before running the final model. The statistical significance and strength of association between independent variables and an outcome variable were measured by a bi-variable logistic regression. Variables with a p-value $< 0.25$ in bi-variable logistic regression were transferred to multi-variable logistic regression model to adjust for confounder effect. Adjusted odds ratio with 95% confidence interval was estimated to assess the strength of the association, and statistical significance was declared at p-value $< 0.05$.

## Ethical consideration

Ethical clearance was obtained from the Institutional Review Board (IRB) of Jimma University; with reference number IHRPD/574/22 and a formal letter of permission were secured from the respective administrative unit of the study setting. The study complies with the declaration

of Helsinki and the study participants were informed of their full right to decline participation from the beginning or to quit participation at any time during the data collection process. Informed written consent was obtained from each participant. After measuring the blood glucose, participants with fasting blood glucose measurement $\geq$ 126 mg/dl were counseled on the control of diabetes mellitus, its complications and instructed to visit the nearby health center for further investigations and treatment. Those who have elevated blood pressure were also counseled on how to control high blood pressure and informed to visit nearby health facility.

## Results

### Socio-demographic and behavioral characteristics of study participants

The study comprised 627 adults, with a response rate of 97.02%. The mean age ($\pm$ SD) of the participants was 37.21($\pm$ 13.5) years. More than half (56.5%) of the participants were female. Regarding educational status, 14.2% of the participants were without formal education (Table 1).

### Clinical characteristics and the prevalence of undiagnosed DM

Above two-thirds, (69.7%) of the participants had BMI below 25kg/m$^2$ and about one-third (28.4%) of participants have high Waist Circumference. About one-fourth (22.3%) of the participants had a family history of diabetes mellitus and a few (9.3%) of participants had blood pressure measurements $\geq$140/90 mmHg (Table 2).

The prevalence of undiagnosed diabetes mellitus was found to be 8.1% (95% CI: 6.1 to 10.6). The study also revealed that the prevalence of pre-diabetes among study participants was, 10.5% (95% CI: 8.2–13.2) with fasting blood glucose levels of 100–125 mg/dl (Fig 2).

### Factors associated with undiagnosed diabetes mellitus

In bi-variable analysis, age, educational status, marital status, occupation, monthly income, level of fruit consumption, level of physical activity, sedentary life, ever-checked blood pressure, family history of DM, body mass index, and current hypertension were found to be associated with the outcome at p<0.25. Finally, on multivariable logistic regression, family history of DM, fewer fruit consumption per week, physical inactivity and having hypertension were significantly associated with UDM at p<0.05.

In multivariable analysis participants who consumed fewer than three servings of fruits per week were 2.64 times more likely to develop undiagnosed DM as compared to participants who consumed more than four servings of fruits per week (AOR = 2.64, 95% CI 1.18, 5.92). Low physical activity was also associated with a 3.4 fold increase of undiagnosed DM as compared to high physical activity (AOR = 3.4, 95% CI: 1.08, 10.66). Compared to people who engage in a sedentary life for less than 4 hours a day, those who engage for more than 4 hours are 3.33 times more likely to have undiagnosed DM (AOR = 3.33, 95% CI: 1.63, 6.79). The odds of developing undiagnosed diabetes mellitus for those who have family history of diabetes mellitus is 2.87 as compared to those who haven't (AOR = 2.87, 95% CI: 1.41, 5.85). Furthermore, being hypertensive was associated with a three-fold increase in undiagnosed DM as compared to being non-hypertensive, (AOR = 2.9, 95% CI: 1.26, 6.69) (Table 3).

## Discussion

The study aimed to assess prevalence of undiagnosed diabetes mellitus and associated factors among adult population in Mizan Aman town, southwestern Ethiopia. Our study found that the prevalence of undiagnosed DM was 8.13%. The finding is comparable to study conducted

**Table 1. Socio-demographic and behavioral characteristics of study participants (n = 627), Mizan-Aman town.**

| Variables | Category | Undiagnosed DM | | Total, N (%) |
|---|---|---|---|---|
| | | Yes, N (%) | No, N (%) | |
| Age (in years) | 18–24 | 4 (3.1) | 123 (96.9) | 127 (20.3) |
| | 25–34 | 8 (5.3) | 144 (94.7) | 152 (24.2) |
| | 35–44 | 15 (8.8) | 156 (91.2) | 171 (27.3) |
| | 45–54 | 12 (12.6) | 83 (87.4) | 95 (15.2) |
| | ≥55 | 12 (14.6) | 70 (85.4) | 82 (13) |
| Sex | Male | 24 (8.8) | 249 (91.2) | 273 (43.5) |
| | Female | 27 (7.6) | 327 (92.4) | 354 (56.5) |
| Educational status | No formal education | 19(21.3) | 70 (78.7) | 89 (14.2) |
| | Primary | 8 (6.4) | 117 (93.6) | 125 (19.9) |
| | Secondary | 12 (4.7) | 244 (95.3) | 256 (40.8) |
| | Diploma & above | 12 (7.6) | 145 (92.4) | 157 (25) |
| Ethnicity | Bench | 15 (6.8) | 204 (93.2) | 219 (34.9) |
| | Kaffaa | 12 (9.6) | 113 (90.4) | 125 (19.9) |
| | Amhara | 9 (12.7) | 62 (87.3) | 71 (11.3) |
| | Oromo | 6 (11.8) | 45 (88.2) | 51 (8.1) |
| | Gurage | 2 (4.2) | 46 (95.8) | 48 (7.7) |
| | Silte | 2 (4.5) | 42 (95.5) | 44 (7) |
| | Others* | 5 (7.2) | 64 (92.8) | 69 (11) |
| Religion | Protestant | 20 (6.6) | 283 (93.4) | 303 (48.3) |
| | Orthodox | 23(10.6) | 195 (89.4) | 218 (34.8) |
| | Muslim | 4 (4.7) | 81 (95.3) | 85 (13.6) |
| | Catholic | 4 (19) | 17 (81) | 21 (3.3) |
| Marital status | Single | 11 (7.2) | 142 (92.8) | 153 (24.4) |
| | Married | 31 (7.4) | 387 (92.6) | 418 (66.7) |
| | Divorce | 3 (9.4) | 29 (90.6) | 32 (5.1) |
| | Widowed | 6 (25) | 18 (75) | 24 (3.8) |
| Occupation | Merchant | 11 (6.5) | 159 (93.5) | 170 (27.1) |
| | Student | 5 (4) | 119 (96) | 124 (19.8) |
| | Housewife | 11 (9.3) | 107 (90.7) | 118 (18.8) |
| | Farmer | 14(14.1) | 85 (85.9) | 99 (15.8) |
| | Employed** | 10 (8.6) | 106 (91.4) | 116 (18.5) |
| Monthly income (ETB) | <1000 | 7 (4.9) | 137 (95.1) | 144 (23) |
| | 1000–1800 | 3 (7.7) | 36 (92.3) | 39 (6.2) |
| | 1801–2400 | 15(11.4) | 117 (88.6) | 132 (21) |
| | >2400 | 26 (8.3) | 286 (91.7) | 312 (49.8) |
| Current smoking | Yes | 9(18) | 41 (82) | 50 (8) |
| | No | 42 (7.3) | 535 (92.7) | 577 (92) |
| Ever chew Khat | Yes | 16 (9.4) | 154 (90.6) | 170 (27.1) |
| | No | 35 (7.7) | 422 (92.3) | 457 (72.9) |
| Ever drink alcohol | Yes | 23 (8.8) | 238 (91.2) | 261 (41.6) |
| | No | 28 (7.7) | 338 (92.3) | 366 (58.4) |
| Fruit consumption (per week) | <3 Servings | 17(17.3) | 81 (82.7) | 98 (15.6) |
| | 3–4 Servings | 7 (7.4) | 88 (92.6) | 95 (15.2) |
| | >4 Servings | 27 (6.2) | 407 (93.8) | 434 (69.2) |

(*Continued*)

**Table 1.** (Continued)

| Variables | Category | Undiagnosed DM | | Total, N (%) |
|---|---|---|---|---|
| | | Yes, N (%) | No, N (%) | |
| Vegetable consumption (per week) | <3 Servings | 9 (6.9) | 122 (93.1) | 131 (20.9) |
| | 3–4 Servings | 8 (7.7) | 96 (92.3) | 104 (16.6) |
| | >4 Servings | 34 (8.7) | 358 (91.3) | 392 (62.5) |
| Physical activity (min/week) | Low (<600) | 28 (11.2) | 223 (88.8) | 251 (40) |
| | Moderate (600–2999) | 18 (7.4) | 226 (92.6) | 244 (38.9) |
| | High (≥3000) | 5 (3.8) | 127 (96.2) | 132 (21) |
| Sedentary life (per day) | <4hr | 29 (9.6) | 479 (90.4) | 508 (81) |
| | ≥4hr | 22 (18.5) | 97 (81.5) | 119 (19) |

Others* = Wolita (28), Dawuroo (19), Hadiya (13), kambata (9), ETB-Ethiopian Birr, Employed**- Government or non-government employed

in Bahir-dar(10.2%), Dire-Dawa(6.2%), Africa(8.84%), Malaysia(8.9%) and Germany(8.2%) [27, 32–36]. However, it is higher than the study results of Jimma, Bishoftu, Koladiba, Iran, Qatar [24, 25, 37–40]. The possible explanation might be difference in sociocultural, healthcare infrastructure, and community awareness related to routine NCD screening. For example; Koladiba's study included rural populations, which are characteristically different from urban populations in terms of risk factors, and some of the studies used small sample sizes [24]. The other possible explanation for the above difference in prevalence is that urban areas in developing countries are currently undergoing a dietary shift towards more unhealthy consumption and junk food, a change in work patterns from heavy work and a Western lifestyle that Need implies more effort in community awareness and education about healthy lifestyles [41].

On the other hand, the prevalence of undiagnosed DM in this study was lower than results of East Gojjam(11.5%), Metu(12.3%), Iraq (11%), Tamil Nadu(11.1%),Urban areas of Egypt (20%) [23, 39, 42–44]. These difference in prevalence might be possibly associated with socio-cultural characteristics, lifestyles, healthcare-seeking behavior, and health infrastructures [45].

**Table 2. Clinical characteristics and laboratory measurements of study participants.**

| Variables | Category | Undiagnosed DM | | Total, N (%) |
|---|---|---|---|---|
| | | Yes, N (%) | No, N (%) | |
| **Ever checked blood pressure** | Yes | 28 (12.4) | 198 (83.6) | 226 (36) |
| | No | 23 (5.7) | 378 (94.3) | 401 (64) |
| **Ever checked blood glucose** | Yes | 9 (9.3) | 88 (90.7) | 97 (15.5) |
| | No | 42 (8) | 488 (92) | 530 (84.5) |
| **History of gestational DM** | Yes | 7 (1.9) | 20 (5.7) | 27 (7.6) |
| | No | 35 (9.9) | 292 (82.5) | 327 (92.4) |
| **Family history of DM** | Yes | 24 (17.1) | 116 (82.9) | 140 (22.3) |
| | No | 27 (5.5) | 460 (94.5) | 487 (77.7) |
| **Current hypertension** | Yes | 15 (25.9) | 43 (74.1) | 58 (9.3) |
| | No | 36 (6.3) | 533 (93.7) | 569 (90.7) |
| **Body mass index in Kg/m2** | <25 | 27 (6.2) | 410 (93.8) | 437 (69.7) |
| | >25 | 24 (12.6) | 166 (87.4) | 190 (30.3) |
| **Waist circumference (in cm)** | Normal | 16 (9) | 162 (91) | 178 (28.4) |
| | High | 16 (9) | 162 (91) | 449 (71.6) |

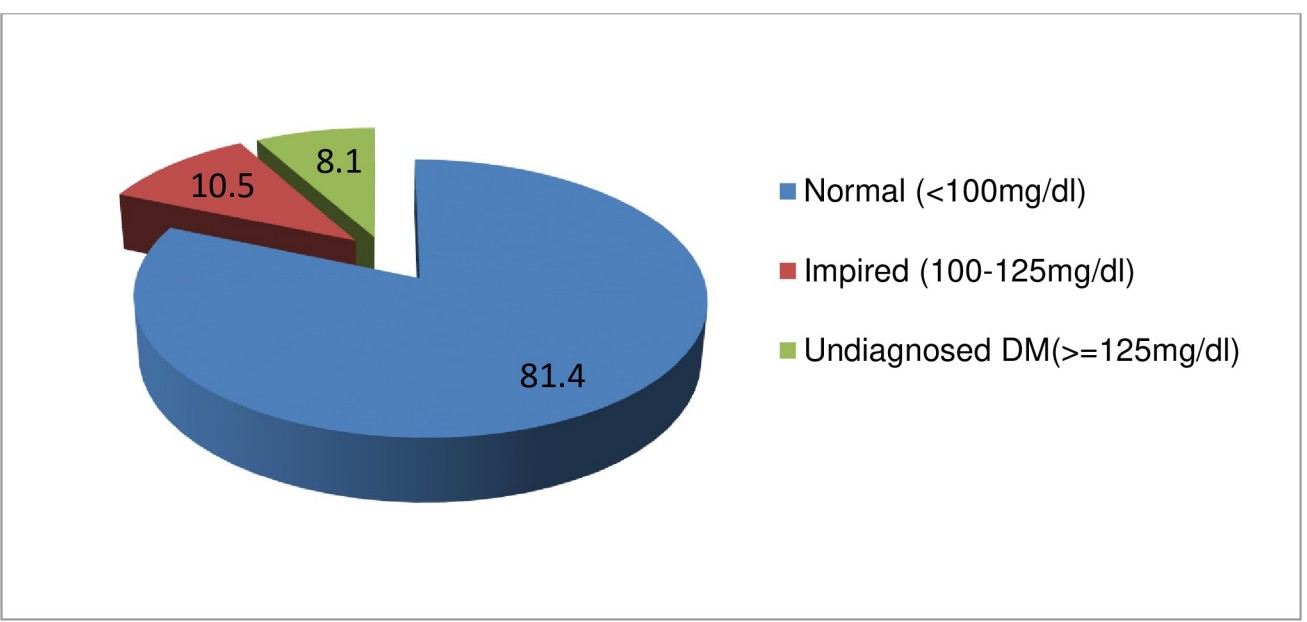

**Fig 2. Fasting blood glucose measurement status of the study participants.**

Concerning the associated factors, the majority of the factors associated with undiagnosed DM in the current study were modifiable risk factors. Undiagnosed diabetes mellitus was more common in participants with low physical activity (< 600 MET-min per week). This was consistent with findings from Qatar, Jimma, South Africa, Kenya and a study in Ethiopia on risk factors for non-communicable diseases [25, 40, 46–48]. The impact of physical activity on reducing the risk of developing undiagnosed diabetes is clear because, during physical activity, glucose uptake into active muscles increases by increasing insulin sensitivity and allowing muscle cells to use glucose efficiently.

Physical activity combined with moderate weight loss has been shown to reduce the risk of type 2 DM by up to 58% in high-risk populations [44]. The study also found that adults who led a sedentary lifestyle were more likely to develop UDM than their peers. The result was consistent with study done in East Gojjam, Jimma town, Mettu town and Brazil respectively [23, 25, 42, 49]. Sedentary lifestyle leading to reduced peripheral insulin-mediated glucose uptake, which secondarily leads to glucose resistance caused by reduced sensitivity of muscle cells, implying that the risk of undiagnosed diabetes mellitus can be minimized by adopting sedentary lifestyle is avoided [49].

This study also found that participants with a family history of diabetes mellitus had higher risk of developing undiagnosed DM than their counterpart, which was consistent with studies done in East Gojjam, Jimma town, Bahir Dar city, Uganda and Northern Sudan respectively [23, 25, 27, 50–54]. According our study, people with hypertension (blood pressure > 140/90 mmHg) are more likely to have undetected diabetes mellitus than participants with normotensive. This result is consistent with study results from East Gojjam, Nekemte town, Bishoftu town, Malaysia and Indonesia respectively [23, 35–37, 55]. The possible explanation is that, in type 2 diabetes mellitus there is body resistance to the insulin, which causes sugar to build up in the blood, leading to damage to the small blood vessels in the body, causing the walls of the blood vessels to become stiff and increasing blood pressure. The other mechanism is that,

**Table 3. Factors associated with undiagnosed diabetes mellitus among adult residents of Mizan-Aman town.**

| Variables | Undiagnosed DM | | COR (95% CI) | AOR (95% CI) | p- value |
|---|---|---|---|---|---|
| | Yes N (%) | No N (%) | | | |
| Age(in years) | | | | | |
| 18–24 | 4 (3.1) | 123 (96.9) | 1 | 1 | |
| 25–34 | 8 (5.3) | 144 (94.7) | 1.71 (0.05, 5.81) | 1.78 (0.4, 7.99) | 0.454 |
| 35–44 | 15 (8.8) | 156 (91.2) | 2.96 (0.96, 9.13) | 3.47 (0.75, 15.97) | 0.110 |
| 45–54 | 12 (12.6) | 83 (87.4) | 4.45 (1.39, 14.26) | 4.25 (0.8, 22.48) | 0.089 |
| ≥55 | 12 (14.6) | 70 (85.4) | 5.27 (1.64, 16) | 3.97 (0.72, 21.83) | 0.113 |
| Educational status | | | | | |
| No formal education | 19 (21.3) | 70 (78.7) | 1 | 1 | |
| Primary | 8 (6.4) | 117 (93.6) | 0.25 (0.11, 0.61) | 0.41 (0.14, 1.22) | 0.109 |
| Secondary | 12 (4.7) | 244 (95.3) | 0.18 (0.08, 0.39) | 0.44 (0.15, 1.33) | 0.147 |
| Diploma & above | 12 (7.6) | 145 (92.4) | 0.30 (0.14, 0.66) | 0.5 (0.12, 2.07) | 0.342 |
| Marital status | | | | | |
| Single | 11 (7.2) | 142 (92.8) | 1 | 1 | |
| Married | 31 (7.4) | 387 (92.6) | 1.03 (0.51, 2.11) | 0.38 (0.14, 1.02) | 0.056 |
| Divorce | 3 (9.4) | 29 (90.6) | 1.34 (0.35, 5.09) | 0.52 (0.1, 2.82) | 0.451 |
| Widowed | 6 (25) | 18 (75) | 4.3 (1.42, 13.04) | 0.37 (0.07, 1.88) | 0.231 |
| Occupation | | | | | |
| Merchant | 11 (6.5) | 159 (93.5) | 0.73(0.3, 1.79) | 0.68 (0.18, 2.53) | 0.561 |
| Student | 5 (4) | 119 (96) | 0.45 (0.15, 1.35) | 0.64 (0.13, 3.13) | 0.580 |
| Housewife | 11 (9.3) | 107 (90.7) | 1.09 (0.15, 1.35) | 1.24 (0.28, 5.6) | 0.778 |
| Farmer | 14 (14.1) | 85 (85.9) | 1.75 (0.74, 4.13) | 0.9 (0.2, 4.06) | 0.885 |
| Employee | 10 (8.6) | 106 (91.4) | 1 | 1 | |
| Monthly income (ETB) | | | | | |
| <1000 | 7 (4.9) | 137 (95.1) | 1 | 1 | |
| 1000–1800 | 3 (7.7) | 36 (92.3) | 1.63 (0.4, 6.6) | 0.71 (0.13, 3.72) | 0.682 |
| 1801–2400 | 15 (11.4) | 117 (88.6) | 2.51 (1, 6.36) | 3.19 (0.97, 10.46) | 0.056 |
| >2400 | 26 (8.3) | 286 (91.7) | 1.78 (0.75, 4.2) | 1.41 (0.45, 4.4) | 0.555 |
| Current smoking | | | | | |
| Yes | 9(18) | 41 (82) | 2.79 (1.27, 6.14) | 1.72 (0.64, 4.57) | 0.281 |
| No | 42 (7.3) | 535 (92.7) | 1 | 1 | |
| Fruit consumption (per week) | | | | | |
| <3 Servings | 17 (17.3) | 81 (82.7) | 3.16 (1.65,6.07) | 2.64 (1.18, 5.92) * | 0.018 |
| 3–4 Servings | 7 (7.4) | 88 (92.6) | 1.2 (0.51, 2.84) | 1.3 (0.49, 3.48) | 0.600 |
| >4 Servings | 27 (6.2) | 407 (93.8) | 1 | 1 | |
| Physical activity | | | | | |
| Low (<600) | 28 (11.2) | 223 (88.8) | 3.19 (1.2, 8.47) | 3.4 (1.08, 10.66) ** | 0.037 |
| Moderate (600–2999) | 18 (7.4) | 226 (92.6) | 2.02 (0.73, 5.58) | 2.61 (0.81, 8.41) | 0.109 |
| High (≥3000) | 5 (3.8) | 127 (96.2) | 1 | 1 | |
| Sedentary life | | | | | |
| ≥4hr per day | 22 (18.5) | 97 (81.5) | 3.75 (2.1, 6.8) | 3.33 (1.63, 6.79) * | 0.001 |
| <4hr per day | 29 (9.6) | 479 (90.4) | 1 | 1 | |
| Ever checked BP | | | | | |
| Yes | 28 (12.4) | 198 (83.6) | 1.18 (0.39, 1.79) | 1.66 (0.84, 3.3) | 0.149 |
| No | 23 (5.7) | 378 (94.3) | 1 | 1 | |
| Family history of DM | | | | | |
| Yes | 24 (17.1) | 116 (82.9) | 3.52 (1.96, 6.34) | 2.87 (1.41, 5.85) * | 0.004 |

*(Continued)*

**Table 3.** (Continued)

| Variables | Undiagnosed DM | | COR (95% CI) | AOR (95% CI) | p- value |
|---|---|---|---|---|---|
| | Yes N (%) | No N (%) | | | |
| No | 27 (5.5) | 460 (94.5) | 1 | 1 | |
| Body mass index | | | | | |
| ≥25 | 24 (12.6) | 166 (87.4) | 2.19 (1.23, 3.92) | 2.01 (0.99, 4.1) | 0.053 |
| <25 | 27 (6.2) | 410 (93.8) | 1 | 1 | |
| Current hypertension | | | | | |
| Yes | 15 (25.9) | 43 (74.1) | 5.16 (2.62, 10.17) | 2.9 (1.26, 6.69) * | 0.012 |
| No | 36 (6.3) | 533 (93.7) | 1 | 1 | |

*Significant at $p < 0.05$, BP-Blood Pressure, 1- Reference, COR-Crude odds ratio, AOR- adjusted odds ratio

insulin resistance-induced hyperinsulinemia and hyperglycemia occur in DM patients, which increase systemic blood pressure [56].

Finally, our study found that participants who consumed fewer than 3 servings of fruits per week were at higher risk of developing undetected diabetes mellitus than those who consumed more than four servings per week which is consistent with a study conducted in china [57]. Fruits and vegetables are fiber-rich in nature which plays an important role in decreasing serum cholesterol and decreasing the release of sugar into the blood which leads to the decrease of blood glucose in the blood by nearly 40% [58]. The World Health Organization (WHO) has recommended that a person should eat 400g of fruit, or five servings of fruit and vegetables, per day [59].

On the contrary, in some studies conducted in another part of Ethiopia, it was found that there was no significant association between fruit consumption and undiagnosed diabetes mellitus, which could be due to the different availability of fresh fruits and vegetables in the two study areas. Portion measurement in developing countries is somewhat challenging [34]. On the other hand, factors such as age, gender, marital status, smoking, glycemic control, and gestational diabetes were not significantly associated with undiagnosed diabetes mellitus, although they were significant in other previously conducted studies. A possible explanation for the difference could be the number of differences between the participants with the characteristics of the difference in the previous study and our study. For example, fewer study participants smoked in our study than in other studies [23].

## Strengths and limitations of the study

This study was community based with reasonable sample size and employed probability sampling and tried to include various variables which were not included in previous studies. However, the study was subject to recall bias and social desirability bias for some behavior-related factors such as smoking, Khat chewing, and alcohol consumption. Furthermore, lipid profile is not included to this study due to resource limitation.

## Conclusion and recommendations

The magnitude of undiagnosed DM in the study area was higher than the national pooled prevalence of UDM in Ethiopia. Physical inactivity, sedentary life, family history of diabetes, hypertension, and low level of fruit consumption were the factors associated with undiagnosed DM. Most of the associated factors in this study were modifiable risk factors and can be solved through creating community awareness and community-based interventional activities.

Efforts should be made by politicians, decision makers and other healthy institutions to implement screening modality and early interventions and finally we suggest for future researcher to include other biochemical measurements like lipid profiles and specifically other blood glucose tests.

## Supporting information

**S1 Checklist. STROBE statement—Checklist of items that should be included in reports of observational studies.**
(DOCX)

**S1 File.**
(DOCX)

**S2 File.**
(DOCX)

**S1 Data.**
(SAV)

## Acknowledgments

The author is grateful to the participants of the study who shared their time to give their genuine responses, data collectors and supervisor.

## Author Contributions

**Conceptualization:** Tsegaye Atrese, Lata Fekadu, Kifle Woldemikael.

**Data curation:** Guta Kune, Abel Shita.

**Formal analysis:** Tsegaye Atrese, Lata Fekadu, Kifle Woldemikael.

**Investigation:** Lata Fekadu, Kifle Woldemikael.

**Methodology:** Tsegaye Atrese, Guta Kune.

**Project administration:** Tsegaye Atrese, Lata Fekadu, Abel Shita.

**Resources:** Tsegaye Atrese, Lata Fekadu, Kifle Woldemikael.

**Software:** Lata Fekadu, Guta Kune, Abel Shita.

**Supervision:** Lata Fekadu, Kifle Woldemikael.

**Validation:** Tsegaye Atrese, Guta Kune, Abel Shita.

**Visualization:** Tsegaye Atrese, Guta Kune, Kifle Woldemikael.

**Writing – original draft:** Tsegaye Atrese, Lata Fekadu, Abel Shita.

**Writing – review & editing:** Tsegaye Atrese, Guta Kune.

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
