## [Decision Letter · Decision Letter 0]

29 Dec 2023

PONE-D-23-12386Prevalence of undiagnosed diabetes mellitus and associated factors among adult residents of Mizan Aman town, Southwest Ethiopia: Community-based cross-sectional studyPLOS ONE

Dear Dr. Kune,

Thank you for submitting your manuscript to PLOS ONE. After careful consideration, we feel that it has merit but does not fully meet PLOS ONE’s publication criteria as it currently stands. Therefore, we invite you to submit a revised version of the manuscript that addresses the points raised during the review process.

We look forward to receiving your revised manuscript.

Kind regards,

Mohammed Hasen Badeso, MPH in Field Epidemiology

Academic Editor

PLOS ONE

Journal Requirements:

3. We note that this data set consists of interview transcripts. Can you please confirm that all participants gave consent for interview transcript to be published?

If they DID provide consent for these transcripts to be published, please also confirm that the transcripts do not contain any potentially identifying information (or let us know if the participants consented to having their personal details published and made publicly available). We consider the following details to be identifying information:

- Names, nicknames, and initials

- Age more specific than round numbers

- GPS coordinates, physical addresses, IP addresses, email addresses

- Information in small sample sizes (e.g. 40 students from X class in X year at X university)

- Specific dates (e.g. visit dates, interview dates)

- ID numbers

Or, if the participants DID NOT provide consent for these transcripts to be published:

- Provide a de-identified version of the data or excerpts of interview responses

- Provide information regarding how these transcripts can be accessed by researchers who meet the criteria for access to confidential data, including:

a) the grounds for restriction

b) the name of the ethics committee, Institutional Review Board, or third-party organization that is imposing sharing restrictions on the data

c) a non-author, institutional point of contact that is able to field data access queries, in the interest of maintaining long-term data accessibility.

d) Any relevant data set names, URLs, DOIs, etc. that an independent researcher would need in order to request your minimal data set.

For further information on sharing data that contains sensitive participant information, please see: https://journals.plos.org/plosone/s/data-availability#loc-human-research-participant-data-and-other-sensitive-data

If there are ethical, legal, or third-party restrictions upon your dataset, you must provide all of the following details (https://journals.plos.org/plosone/s/data-availability#loc-acceptable-data-access-restrictions):

a. A complete description of the dataset

b. The nature of the restrictions upon the data (ethical, legal, or owned by a third party) and the reasoning behind them

c The full name of the body imposing the restrictions upon your dataset (ethics committee, institution, data access committee, etc)

4. If the data are owned by a third party, confirmation of whether the authors received any special privileges in accessing the data that other researchers would not have

5. Direct, non-author contact information (preferably email) for the body imposing the restrictions upon the data, to which data access requests can be sent

Reviewers' comments:

Reviewer's Responses to Questions

**Comments to the Author**

1. Is the manuscript technically sound, and do the data support the conclusions?

Reviewer #1: Yes

Reviewer #2: No

Reviewer #3: Partly

2. Has the statistical analysis been performed appropriately and rigorously? 

Reviewer #1: Yes

Reviewer #2: No

Reviewer #3: Yes

3. Have the authors made all data underlying the findings in their manuscript fully available?

Reviewer #1: Yes

Reviewer #2: Yes

Reviewer #3: Yes

4. Is the manuscript presented in an intelligible fashion and written in standard English?

Reviewer #1: No

Reviewer #2: No

Reviewer #3: No

5. Review Comments to the Author

Reviewer #1: Prevalence of undiagnosed diabetes mellitus and associated factors among adult residents of Mizan Aman town, Southwest Ethiopia: Community-based cross-sectional study

Thank you very much for giving me the opportunity to review this well written Manuscript

From the introduction part of the abstract section: You should show why you are going to conduct this study? Is it a public health concern?

What is WHO STEP wise method? Is it a tool? If so, determine its validity and reliability? The information present in the abstract should be present the main document. But this information/statement is not found in the main document.

Why did you use fasting blood glucose level to determine whether the individual has diabetes mellitus or not? Why did you fail to use either random blood glucose or OGTT?

Did you think that only single measurement of blood glucose is sufficient to declare whether the patient has diabetes mellitus or not? How do you exclude those risk factors which may elevate blood glucose level?

Did you think that only single measurement of blood pressure is sufficient to declare whether the patient has hypertension or not? How do you exclude those risk factors which may elevate blood pressure level?

Could you explain the entire procedure to measure random blood glucose level? How do you know, if the person eats breakfast and he said that he is fasting for more than 8 hour? How do you detect such types of error/bias? How do you detect measurement error? Do you know the sensitivity and specificity of the measurement tool?

Why do you focus on the prevalence of undiagnosed DM among adult people? Did you think that, it is not common in pediatrics age group? Or DM is not a concern for pediatrics population?

As you stated in the introduction section, despite different strategies were undertaken to reduce the incidence of DM, the majority of individuals come to healthcare after the disease has been progressed and multiple organ damage has occurred. So, it is better to study why people fail to measure/to be measured their blood glucose level regularly? As per my understanding, the effect of undiagnosed DM is not mentioned.

Why you are going to conduct this study, since it is studied and published in the different regions and corners of the country? You just tried to justify under the last paragraph of the introduction section but it is not persuasive as intended.

What is your inclusion criterion?

Why did you exclude pregnant women and adults who were taking beta blockers within two weeks of data collection?

Why did you use 3% margin of error, and 1.5 design effect?

The sampling procedure is not well explained. So it needs revision and it is recommended to support using diagrammatic presentation. In the case of households with no eligible participant at the time of visit, the next nearby household was selected as a substitute why you fail to revisit?

The tool is adapted from Ethiopia's steps report on risk factors for non-communicable diseases and the prevalence of selected NCDs (25)……it is not similar with the idea stated under reference 25. So. It needs revision. Similarly see reference number 28

Could you explain about the specificity and sensitivity of CareSens TM N Eco digital glucometer?

Why did you select those variables with a p-value < 0.25 in bi-variable logistic regression transferred to multi-variable logistic regression model to adjust for confounder effect? What is your evidence and justification???

What is the declaration of Helsinki? How did you get blood sample? Did the study participants were volunteer? Could you tell me any event you experienced while collecting blood from the people? Did they ask any compensation for the harm? Please explain each ethical standard.

How do you classify monthly income in to these categories? Did you have any evidence? If so what is your evidence?

In bi-variable analysis, 12 variables were identified as statistically significant variable……..it is not necessary to mention the number rather it is enough to mention the variable solely.

Why did you fail to interpret each and every variable’s which had statistically significant association with the outcome variable? Please interpret each variable

Some terminology needs operational definition. Eg sedentary life

In the discussion section, you write reference number rather than writing the study setting/area. Please write the specific area/town/country and some justification given for the possible association is not persuasive. So it needs revision.

Your strength of the study is not ideal

Reviewer #2: � There are grammatical and syntax errors throughout the manuscript. The manuscript needs extensive review by the author and should also be checked by a professional speaker of English.

Please write abbreviations in full at the first time of use.

The objectives and rationale of the study are not clearly stated. Why not include impaired fasting sugar level, or pre-DM, and its risk factors in your study? Previously, almost similar studies had been conducted in the same study area, so what is the significance of doing this study?

Abstract:

Background: In the background part, include the problem or identified gap that you want to fill.

Methods: Please include the laboratory method that you used to determine the fasting blood sugar level briefly.

Avoid abbreviations from the abstract.

Introduction: Add information about the complications and risk factors of DM. Some literature is not up-to-date and needs to be reviewed properly.

Methods and materials:

The sampling technique is not well stated. How many households were in each cluster? How was the selection made? Please include the detailed sampling frame.

How can you be sure that patients were fasting overnight? Is there any system used to convince them and trust them?

Add relevant citations for important statements.

Quality control measures are inadequate. What quality control measures were you using for the CareSensTM N Eco digital glucometer to ensure the method was safe?

Please operationally define physical activity, sedentary life, hypertension, undiagnosed DM, and so on.

Ethical consideration: What was done for those with raised blood pressure?

Results:

The presentation of the results is poor in terms of sentence form, coherence, outcome measurement, and lots of grammatical errors. The authors are unable to provide the most important result in a short and precise manner. Furthermore, the results are simply summarized in the tables without sufficient explanation. In general, the results section needs major modification.

The authors did not include some important variables, such as lipid profile, in their study.

Diagrams and figures are missing. Add figures and diagrams.

The study has many more limitations, but the authors did not include them as the limitations of the study.

Reviewer #3: General comment

The author tried to estimate the prevalence of undiagnosed DM and its predictors among residents of Mizan Aman town. However, the predictive variables such as hypertension, family Hx, sedentary life, and physical activity are the very well known risk factors for DM. So what value this research could add to the scientific world. This is the major gap of the study.

Introduction

Most of the sentences in this section are not paraphrased by the authors own word. For instance the first sentence is directly copied from somewhere. So the authors have to re-write the sentences by their own word.

Methods

Why adults taking beta blockers are excluded from the study?

The tools the author used for dietary questions should be described in detail

Some terms like ever drinker/chewer, current chewer/drinker, and physical activity should be operationally defined.

Result

In table 2, there is a variable “history of gestational diabetes”. I think this variable should be only for female participants. But the authors used it for both males and females. How?

In table 3, the P-value for COR should be stated.

Discussion and conclusion

Being community based study by itself cannot be the strength of your study. There are an ample community based studies conducted on undiagnosed DM among residents of Ethiopia.

The conclusion and recommendation part is very weak. There is no direction toward future studies, emphasis on what needs to be done, or what is missing.

6. PLOS authors have the option to publish the peer review history of their article (what does this mean?). If published, this will include your full peer review and any attached files.

Reviewer #1: No

Reviewer #2: No

Reviewer #3: **Yes: **Misganaw Asmamaw Mengstie

---

## [Author Response · Author response to Decision Letter 0]

13 Feb 2024

I very grateful for reviewers and editors valuable comments.

---

## [Editor Report · Decision Letter 1]

28 Mar 2024

Prevalence of undiagnosed diabetes mellitus and associated factors among adult residents of Mizan Aman town, Southwest Ethiopia: Community-based cross-sectional study

PONE-D-23-12386R1

Dear Author(s),

We’re pleased to inform you that your manuscript has been judged scientifically suitable for publication and will be formally accepted for publication once it meets all outstanding technical requirements.

Kind regards,

Mohammed Hasen Badeso, MPH in Field Epidemiology

Academic Editor

PLOS ONE